# GIGYF2: A Multifunctional Regulator at the Crossroads of Gene Expression, mRNA Surveillance, and Human Disease

**DOI:** 10.3390/cells14131032

**Published:** 2025-07-05

**Authors:** Chen-Shuo Zhao, Shu-Han Liu, Zheng-Yang Li, Jia-Yue Chen, Xiang-Yang Xiong

**Affiliations:** 1The MOE Basic Research and Innovation Center for the Targeted Therapeutics of Solid Tumors, School of Basic Medical Sciences, Jiangxi Medical College, Nanchang University, Nanchang 330031, China; 4203122067@email.ncu.edu.cn (C.-S.Z.); 4203121093@email.ncu.edu.cn (S.-H.L.); 4203121455@email.ncu.edu.cn (J.-Y.C.); 2HuanKui Academy, Nanchang University, Nanchang 330031, China

**Keywords:** GIGYF2, GYF domain, mRNA surveillance, translational repression, mRNA degradation

## Abstract

GIGYF2 (Grb10-interacting GYF protein 2) functions as a versatile adaptor protein that regulates gene expression at various levels. At the transcriptional level, GIGYF2 facilitates VCP/p97-mediated extraction of ubiquitylated Rpb1 from stalled RNA polymerase II complexes during DNA damage response. In mRNA surveillance, GIGYF2 participates in ribosome collision-induced quality control, nonsense-mediated decay, no-go decay, and non-stop decay pathways. Furthermore, GIGYF2 interacts with key factors including 4EHP, TTP, CCR4-NOT, DDX6, ZNF598, and TNRC6A to mediate translational repression and mRNA degradation. Additionally, dysregulation of GIGYF2 has been implicated in various pathological conditions, including metabolic diseases, vascular aging, viral infections, and neurodegenerative disorders. This review summarizes the structural and functional characteristics of GIGYF2, highlighting its importance in transcriptional regulation, mRNA surveillance, translational inhibition, and mRNA degradation, while also elucidating its potential as a therapeutic target for disease treatment.

## 1. Introduction

The GYF domain represents a highly conserved, versatile adaptor module that specifically recognizes proline-rich sequences (PRS) [1]. Initially identified in CD2BP2 as the structural element mediating interaction with PPPGHR amino acid repeats within the T-cell adhesion molecule CD2 [2], this domain derives its nomenclature from the signature glycine-tyrosine-phenylalanine tripeptide sequence [1]. Comprising approximately 60 amino acid residues, the GYF domain is characterized by a conserved signature motif GPF-X4-[M/V/I]-X2-W-X3-GYF that assembles into a distinctive bulge-helix-bulge tertiary structure [3]. The GYF domain family segregates into two distinct subfamilies: CD2BP2-type and SMY2-type [1,2,4]. The CD2BP2-type, first characterized in T-cell adhesion molecule CD2 binding protein 2, is distinguished by a tryptophan residue at position 22 and a longer β1-β2 loop region [5,6]. In contrast, the SMY2-type, named after the yeast suppressor of myosin 2 protein, exhibits an aspartic acid at the corresponding position 22, and the domain prefers ligands with the consensus motif PPGF (where f = hydrophobic amino acid, except for tryptophan) [7].

GYF domain-containing proteins participate in diverse cellular processes, including proteome-wide protein–protein interactions, translational regulation, mRNA splicing, surveillance mechanisms, ubiquitin conjugation, signal transduction pathways, and regulation of immunoproteasome [7,8,9,10,11,12,13,14,15,16].

The human genome encodes two principal SMY2-type GYF domain-containing proteins: GIGYF1 and GIGYF2. GIGYF2 (Grb10-interacting GYF protein 2) differs structurally from GIGYF1 by containing glutamic acid-rich and glutamine/proline-glutamine-rich C-terminal sequences, whereas GIGYF1 features a distinctive polyproline region [17]. However, given that the functional mechanisms and biological significance of GIGYF1 remain poorly characterized, this review focuses on elucidating the multifaceted roles of GIGYF2 in regulating various stages of gene expression.

GIGYF2 is encoded by the *GIGYF2* gene, which belongs to the GIGYF protein family. GIGYF2 was initially discovered through yeast two-hybrid screening, where researchers used the N-terminal region of Grb10 (growth factor receptor-bound protein 10) as bait and identified novel interacting proteins, including GIGYF1 and GIGYF2, from a mouse fat cell cDNA library [18]. At the organ level, GIGYF2 is widely expressed throughout the body, with particularly high expression in the liver, pancreas, brain, lung, kidney, and spleen [19]. At the subcellular level, GIGYF2 exhibits dual localization patterns depending on cellular conditions. Under normal physiological conditions, GIGYF2 associates with the endoplasmic reticulum (ER) and Golgi apparatus, where it colocalizes with COPII vesicle proteins (particularly Sec31A) involved in protein transport [7]. However, under cellular stress conditions such as arsenite exposure, GIGYF2 redistributes to cytoplasmic stress granules, which contain translationally repressed mRNAs and RNA-binding proteins like TIA-1 [7].

Despite their structural similarities, GIGYF1 and GIGYF2 operate through distinct functional mechanisms. GIGYF1 primarily represses translation by strongly interacting with eIF3 complex subunits (particularly eIF3L, eIF3E, eIF3D, and eIF3G) through its C-terminal region, disrupting the critical eIF3-eIF4G1 interaction. In contrast, GIGYF2 relies approximately 50% on its interaction with 4EHP (eIF4E2) for translational repression [17,20].

The understanding of GIGYF2 function has undergone remarkable expansion in recent years. Initially identified as a binding partner of GRB10 and implicated in growth factor receptor signaling and insulin/IGF1 pathways [19], subsequent studies have revealed its multifaceted roles. Current research highlights GIGYF2’s involvement in neurodevelopment and neurodegenerative diseases [21,22], mRNA regulation through interactions with RNA-binding proteins (AGO2, DDX6) [23,24], direct participation in mRNA degradation as an RNA-binding protein [25], and potential tumor suppressor functions [26]. Additionally, GIGYF2 has been linked to autophagy regulation [12] and metabolic processes affecting glucose and lipid metabolism [27]. In this review, we summarized the role of GIGYF2 in various stages of gene expression regulation.

## 2. The Structure of GIGYF2

The human *GIGYF2* gene is located on chromosome 2q37.1, spanning 163,275 base pairs and comprising 35 exons. The GIGYF2 protein consists of 1299 amino acids with a theoretical molecular weight of 150,070 Da. GIGYF2 contains three distinct domains that contribute to its repressive activity: an N-terminal domain (NTD, amino acids 1-532), a GYF domain (amino acids 533-596), and a C-terminal domain (CTD, amino acids 606-1299) [25]. The NTD includes binding domains for both 4EHP and DDX6 [23,28], while the CTD contains the NSP2 binding domain [29,30,31] (Figure 1A).

### 2.1. N-Terminal Domain

#### 2.1.1. 4EHP-Binding Motif

GIGYF2 binds to 4EHP to form a complex that inhibits translation initiation by competing with eIF4E to bind to the 5’ cap structure of target mRNAs, thereby preventing the assembly of the translation initiation complex [20,32,33]. Although 4EHP exhibits lower affinity for the 5’ cap compared to eIF4E [33,34], this competition is likely enhanced by GIGYF2, which stabilizes 4EHP and may increase its cap-binding capability [28].

The 4EHP-binding region (4EHP-BR) of GIGYF2 employs a tripartite binding mechanism consisting of three distinct sequence elements that work cooperatively to achieve high-affinity and specific binding to 4EHP [28] (Table 1) (Figure 1B). First, the canonical motif (YXYX4LΦ consensus sequence, residues 41–49) forms the primary interaction by binding to the dorsal surface of 4EHP in a helical conformation, similar to how 4E-BPs bind to eIF4E. Second, the noncanonical motif (located ~12 residues C-terminal to the canonical motif) engages the lateral surface of 4EHP through hydrophobic interactions, providing additional binding stability. Third, and most importantly for 4EHP specificity, the auxiliary sequences (A1–A3) extend the binding interface and contact 4EHP-specific residues that are not conserved in eIF4E.

The auxiliary sequences can be subdivided into three functional elements: (1) Auxiliary motif 1 (A1), which contains the invariant PLAL motif and is connected to the noncanonical motif via auxiliary linker 1 (a-L1); (2) Auxiliary motif 2 (A2), which forms auxiliary helix α2; and (3) Auxiliary motif 3 (A3), which adopts auxiliary helix α3 and is positioned near the 4EHP cap-binding pocket (Figure 1C). These auxiliary motifs are interconnected by conserved linkers a-L2 and a-L3, with the a-L3 linker containing a critical VNS sequence that interacts with 4EHP-specific residues [28]. Importantly, the auxiliary sequences increase GIGYF2’s affinity for 4EHP by 30–40 fold compared to peptides lacking these sequences, and they interact specifically with 4EHP residues (R103, R140, E149) that differ from eIF4E, thereby conferring selectivity for 4EHP over eIF4E.

Within the asymmetric unit of 4EHP-GIGYF2 crystals, the GIGYF2 canonical and auxiliary motifs constitute part of a large interface (1008 Å^2^), which mediates the association of two adjacent 4EHP-GIGYF2 complexes. The putative dimer interface is stabilized by 4EHP-specific residues R202, M161, and Q159, and GIGYF2 residues E46 and E47 [28]. This complex dimerization potentially increases the local concentration of repressor complexes on target mRNAs, thereby amplifying their regulatory efficacy.

#### 2.1.2. DDX6-Binding Motif

GIGYF2 interacts with the RNA-dependent ATPase DDX6/Me31B through a conserved Me31B/DDX6-binding motif (MBM) to mediate translational repression [23,35,36]. Structurally, the MBM employs a bipartite binding mechanism: the N-terminal PEW (Pro-Glu-Trp) sequence forms a short coil, with the tryptophan residue (W349 in Drosophila GIGYF, equivalent to W221 in 4E-T) inserting into the W pocket between helices α10 and α11 of Me31B (Table 1). The PEW region is connected via a flexible linker to a C-terminal β-hairpin structure containing a unique “split” FDx_4_F motif (F361, D362, and F367 in Drosophila GIGYF) that engages the FDF pocket of Me31B.

**Table 1 cells-14-01032-t001:** Key residues in GIGYF2 protein interactions with 4EHP (human), Me31B/DDX6 (Dm) and TNRC6A (Human).

Interaction Proteins	GIGYF2 Components	GIGYF2 Residues	Interaction Partners	References
4EHP (Human)	Canonical Motif	Y41	P55	[28]
		M48/L49/Y43	4EHP dorsal surface	
		F52	W95/P78	
	Noncanonical Loop	F67	Y64/K83/I85	
		I70	Y64	
		I70/Q72	H100/V102	
	Linker	I58/D55	H100	
		P59	F97	
	A1	L79/A80	E149	
		L79/V82	R146	
		P77	R103	
	A3	S98	R138	
		S98-OH	E177	
		V101/L102	R138/I211	
Me31B/DDX6 (Dm)	MBM Motif	P347/E348 /W349	V283/L310/L311 /F370	[23]
		F361/F367	A275/H284/C285 /L289/I421	
		D362	H368	
		G365/F367	F276	
		F361	K423	
TNRC6A (Human)	GYF Domain	Y538/F549 /W557/Y562	P1481/P1482	[37]

This bipartite binding strategy distinguishes GIGYF from other DDX6-interacting proteins by combining structural elements reminiscent of the 4E-T PEW motif with its specialized “split” FDx_4_F configuration. Critically, the molecular basis for selective ternary complex formation lies in the differential binding partner requirements. NOT1, the scaffold protein of the CCR4-NOT deadenylase complex, binds to a distinct surface on the DDX6 RecA2 domain that is spatially adjacent to, but non-overlapping with, the FDF pocket utilized by other DDX6 partners [23].

The capacity for ternary complex assembly is determined by the electrostatic compatibility between binding partners and NOT1. Specifically, EDC3 and PatL1 contain negatively charged residues N-terminal to their FDF motifs, which generate unfavorable electrostatic repulsions with NOT1, precluding ternary complex formation. In contrast, both 4E-T and GIGYF feature polar rather than negatively charged residues preceding their respective binding motifs (IEL and FDx_4_F), thereby permitting the formation of stable ternary complexes: 4E-T-DDX6-NOT1 and GIGYF-DDX6-NOT1 [23]. This structural compatibility enables GIGYF to orchestrate translational repression of target mRNAs through coordinated recruitment of both the DDX6 helicase and the CCR4-NOT deadenylation machinery.

### 2.2. GYF Domain: Interactions with ZNF598/TTP/TNRC6A

The GYF domain of GIGYF2 (residues 529–597) is a highly conserved structural unit comprising a four-stranded antiparallel β-sheet with two α-helices packed onto one face [37]. Named after its characteristic glycine-tyrosine-phenylalanine motif within the larger GPF-X4-[M/V/I]-X2-W-X3-GYF signature sequence, this domain forms a bulge-helix-bulge structural element generating a hydrophobic ligand-binding surface [1]. Functionally, the GYF domain can be divided into two distinct regions: the N-terminal portion (β1, β2, α1, and α1-β3 loop) constitutes the essential ligand-binding interface for GIGYF2-mediated translational repression, while the C-terminal region shows greater variability across proteins [3]. Notably, the C-terminal phenylalanine residue (F592 in GIGYF2), termed the “Phe plug”, is highly conserved and inserts into a hydrophobic cavity between the α1 and α2 helices and the β-sheet’s inner surface, significantly contributing to domain stability and forming part of the hydrophobic core essential for the domain’s structural integrity [37].

The GYF domain of GIGYF2 mediates specific protein interactions by recognizing PPGΦ motifs, where P represents proline, G represents glycine, and Φ denotes a hydrophobic amino acid [6,7]. This recognition depends on a conserved binding groove formed by residues Y538, F549, W557, Y562, F563, and L567, which accommodates the PPII helical conformation adopted by proline residues in binding partners [37]. The conserved glycine in these motifs is critical as it prevents steric clashes with D540 and Q546 residues while enabling the hydrophobic Φ residue to insert into a specific cavity [7,37]. GIGYF2’s key binding partners—TNRC6A, TTP, and ZNF598—all utilize this interaction mechanism but display important molecular distinctions that confer functional specificity. TNRC6A contains PPGL motifs where leucine occupies the Φ position, precisely fitting into the Smy2 subclass-specific cavity of GIGYF2’s GYF domain [37,38] (Table 1) (Figure 1D). In contrast, both TTP and ZNF598 feature PPPPGF motifs with phenylalanine at the Φ position [7]. ZNF598 contains three polyproline stretches that are structurally similar to those in TTP [38]. Importantly, these interactions are predominantly transient and context-dependent rather than constitutive associations. TNRC6A-GIGYF2 interactions are specifically recruited during miRNA-mediated gene silencing when target mRNAs are bound by the miRISC complex [39]. TTP-GIGYF2 associations are primarily induced under inflammatory conditions when AU-rich element-containing mRNAs require translational repression [38]. ZNF598-GIGYF2 interactions are triggered during ribosome collision events and translational quality control responses [15,40]. Additionally, these GYF domain-dependent interactions can be negatively modulated by stress-induced GIGYF2 phosphorylation, which recruits 14-3-3 proteins and disrupts binding partner associations [41]. Notably, GIGYF2 targeting of mRNAs occurs through dual mechanisms: indirect recruitment via protein–protein interactions with RNA-binding partners (such as TTP recognizing AU-rich elements or TNRC6A in miRISC complexes) and direct RNA binding through its intrinsic RNA-binding regions located in both the N-terminal (residues 1–532) and C-terminal (residues 606–1299) domains [25]. In the context of ribosome collision events, GIGYF2 recruitment is primarily mediated through ZNF598 recognition of collided ribosome surfaces rather than through direct RNA binding or nascent peptide recognition [15,40]. This molecular architecture enables GIGYF proteins to participate in context-specific translational control through a common binding mechanism with partner-specific variations.

### 2.3. C-Terminal Domain Interactions: Cooperative Roles of Multiple Structural Elements

Studies have demonstrated that molecular interactions involving GIGYF2 frequently necessitate the coordinated participation of multiple structural domains. First, GIGYF2 is involved in the interaction with NSP2 through diverse regions [30,31]. The C-terminal region spanning residues 743–1085 of GIGYF2, along with 4EHP, functions as a critical contact interface for the N-terminal region of NSP2 (residues 1–350), which contains a highly conserved zinc finger domain. Additionally, GIGYF2’s N-terminal domain (residues 1–267), containing the 4EHP-binding motif, also participates in this interaction [31]. Second, GIGYF2 establishes multiple interfaces with the CCR4-NOT complex through RNA-independent mechanisms. The three major domains of GIGYF2 exhibit differential binding preferences with CCR4-NOT subunits—the GYF domain (when attached to the NTD) preferentially interacts with CNOT1 and CNOT7, while the CTD and NTD + GYF regions show stronger affinity for CNOT9 [25]. Third, GIGYF2 directly engages with endogenous mRNAs through two distinct RNA-binding regions: the N-terminal domain (amino acids 1–532) and the C-terminal domain (beyond amino acid 606) [25]. Fourth, GIGYF2 undergoes phosphorylation at two evolutionarily conserved serine residues, S157 and S638, which flank the GYF domain [41]. This phosphorylation is catalyzed by MAP kinase-activated protein kinase 2 (MK2), a downstream effector of the p38 MAPK signaling cascade that is activated during cellular stress responses [42]. The resultant phosphorylation generates binding sites for 14-3-3 proteins, which negatively affects GYF domain-dependent protein interactions [41].

## 3. Transcriptional Regulation by GIGYF2

When DNA sustains damage from agents such as UV radiation and conventional repair mechanisms are insufficient, the cellular “last resort” pathway is activated as a critical transcription stress response [43]. This pathway involves the selective extraction and degradation of the Rpb1 subunit from stalled RNA polymerase II (RNAPII) complexes in chromatin [43,44,45], which may contribute to the recruitment and activity of repair factors involved in the global genome nucleotide excision repair and transcription-coupled nucleotide excision repair [46,47].

In *Saccharomyces cerevisiae*, the GIGYF homolog Smy2 functions as a key regulatory factor in the evolutionarily conserved “last resort” pathway through a multi-step interaction mechanism (Figure 2A). The pathway begins with mono-ubiquitylation of Rpb1 (the largest subunit of RNA polymerase II) by the ubiquitin ligase Rsp5 [48]. Concurrently, Smy2 collaborates with Ubx1 and Cdc48 (cell division cycle protein 48) to facilitate efficient proteasome-mediated processing of Def1 into pr-Def1 [49]. The accumulation of pr-Def1 promotes the recruitment of the Elc1-Ela1-Cul3 ligase complex to mono-ubiquitylated Rpb1, catalyzing its subsequent polyubiquitylation [50]. Following this, Smy2 works in concert with Cdc48 and its co-factors Ubx4/5 to extract polyubiquitylated Rpb1 from damage-stalled RNAPII complexes within chromatin [45,49]. The extracted ubiquitylated Rpb1 is then targeted for degradation by the 26S proteasome [51]. Notably, immunoprecipitation experiments have revealed that significant co-precipitation of Cdc48 and Rpb1 occurs only in the presence of Smy2 expression, highlighting the essential bridging function of Smy2 in mediating this critical interaction [49].

In human cells, GIGYF1/2 proteins serve as key regulatory factors that interact with VCP (the mammalian homolog of yeast Cdc48), specifically facilitating the extraction and proteasomal degradation of ubiquitylated POLR2A (the human homolog of yeast Rpb1) [47,52] from stalled transcription complexes [49] (Figure 2B). Sophisticated biochemical fractionation experiments have demonstrated that GIGYF2 maintains substantial chromatin association, which becomes markedly enhanced following UV-induced DNA damage, highlighting its direct involvement in the transcription stress response [49]. Recent research has identified UBAP2/UBAP2L as the human orthologs of yeast Def1, establishing their critical role in recruiting the Elongin-Cul5 ubiquitin ligase complex to sites of DNA damage [47]. This recruitment follows a highly coordinated two-step ubiquitylation mechanism: NEDD4 (the human ortholog of yeast Rsp5) acts as the initial E3 ubiquitin ligase [53], adding a single ubiquitin moiety to POLR2A, after which the Elongin-Cul5 ubiquitin ligase complex builds K48-linked polyubiquitin chains on this mono-ubiquitinated substrate [52,54,55,56]. These K48-linked chains effectively mark POLR2A for proteasomal degradation, representing a critical “last resort” pathway in the cellular response to transcription-blocking DNA damage [56].

These molecular mechanisms collectively establish GIGYF1/2 and Smy2 as regulators of the transcriptional stress response and DNA damage repair pathway. Their function as previously overlooked modulators of Cdc48/VCP activity extends beyond transcription stress to multiple cellular processes, representing an important advancement in our understanding of cellular quality control systems and stress response mechanisms.

## 4. GIGYF2’s Role in mRNA Surveillance

During the translation of aberrant transcripts, suboptimal codons with limited cognate tRNA availability significantly reduce ribosomal transit rates [57]. This programmed deceleration triggers ribosomal collisions along the mRNA transcript [58,59,60]. The collision events function as molecular signaling nodes, effectively coupling translation elongation kinetics to mRNA surveillance. The mRNA surveillance mechanism enables cells to identify and target aberrant mRNAs for degradation, preventing the translation of defective transcripts through three key pathways: nonsense-mediated decay (NMD), no-go decay (NGD), and non-stop decay (NSD) [61,62]. NMD targets transcripts with premature termination codons [63]. NGD recognizes and degrades transcripts on which ribosomes stall within the CDS due to the presence of inhibitory stem-loop structures or in stretches of rare codons [64,65]. NSD handles mRNAs lacking stop codons [66,67]. Working in concert with these systems, ribosome-associated quality control (RQC) resolves stalled translation complexes by rescuing ribosomes and degrading incomplete nascent polypeptides [68]. Recent studies have shown that GIGYF2 plays an important role in mRNA surveillance by forming a complex with 4EHP that binds to the 5’ cap of mRNAs, thus preventing ribosome loading to aberrant transcripts. This complex participates in multiple mRNA surveillance pathways and prevents the continued translation of defective mRNAs, particularly those encoding secretory proteins, membrane proteins, and tubulin subunits [15].

### 4.1. Transient Collision-Induced ISR Pathway: GIGYF2’s Early Response to Ribosomal Stalling

When two ribosomes collide, EDF1 (endothelial differentiation-related factor 1) is the first responder due to its high cellular abundance [40,69,70]. During transient collisions, EDF1’s engagement triggers a series of coordinated responses. Initially, EDF1 utilizes its conserved KKW motif and alpha-helical segment to clamp the mRNA in a headlock-like arrangement [71], stabilizing collided ribosomes and preventing frameshifting. Through RACK1-dependent and ZNF598-independent mechanisms, EDF1 recruits and stabilizes the GIGYF2-4EHP complex to collision sites [40] (Figure 3A). The GIGYF2-4EHP complex implements translational inhibition via 4EHP-mediated sequestration of the 5’ cap structure, effectively reducing ribosome density along the mRNA [20,71]. Notably, this regulatory mechanism requires both ribosome stalling and intact cap-binding functionality [15].

Resolution of the collision event proceeds through the activity of the eEF1A·GTP·aminoacyl-tRNA complex, which delivers the appropriate charged tRNA to the ribosome A-site, thereby facilitating the resumption of translational elongation [40]. Beyond its immediate role in collision resolution, EDF1 demonstrates additional regulatory functions, particularly in the JUN-mediated transcriptional response to ribosomal collisions [40]. Simultaneously with these events, GCN2 senses ribosome collision events, triggering phosphorylation of eIF2α and activation of the Integrated Stress Response (ISR) pathway [72,73]. The phosphorylated eIF2α inhibits eIF2B activity and consequently reduces global translation [74,75]. However, this translational reprogramming selectively preserves the expression of a small number of stress response mRNAs, including GCN4 and ATF4 [76].

### 4.2. Persistent Collision-Activated RQC and RSR Pathways: GIGYF2’s Role in Sustained Translational Stress

Ribosome collisions occur when trailing ribosomes encounter stalled leading ones during translation, primarily due to genetic mutations, mRNA processing errors, insufficient charged tRNAs, and poly(A) tracts created by premature polyadenylation [77,78,79]. This creates distinctive 40S-40S interfaces that serve as recognition sites for quality control mechanisms [64,80]. EDF1, a highly abundant protein, acts as the primary sensor of these collisions, binding to the collided ribosomes and stabilizing the recruitment of the ZNF598-GIGYF2-4EHP complex [40]. In persistent collisions, E3 ubiquitin-protein ligase ZNF598 recognizes collided di-ribosome structures on non-stop mRNA substrates [81] by specifically binding to target sites at the 40S-40S interface [64,80], particularly when a trailing ribosome catches up to a stalled one. The GIGYF2-4EHP complex inhibits translation initiation by competing with eIF4E for the mRNA 5’ cap, providing an initial response to reduce ribosome density on problematic transcripts (Figure 3B).

If collisions persist despite this initial response, ZNF598 ubiquitinates several 40S ribosomal proteins (primarily eS10) over approximately 2–5 min to accumulate to significant levels under near-physiological conditions [82]. This time delay ensures that ZNF598-dependent quality control is only initiated when persistent ribosome collisions cannot be resolved through conventional translation dynamics [40]. The ubiquitination of eS10 serves as a signal for the ASC-1 complex (ASCC), which contains the ASCC3 helicase, to recognize and act on the stalled ribosome. ASCC then directly dissociates the lead ribosome into 40S and 60S subunits, irreversibly aborting translation and triggering ribosome quality control (RQC) [62,80,83,84,85,86]. Following ribosome separation, the resultant 60S subunit-peptidyl-tRNA complex undergoes processing via the RQC machinery, comprising Listerin (an E3 ubiquitin ligase), NEMF, and additional factors that facilitate appropriate management of incomplete nascent peptides [87]. This mechanism prevents the accumulation of potentially cytotoxic protein fragments within the cellular environment. Upon resolution of the stalled lead ribosome, trailing ribosomes are liberated from their collision state and can resume translational elongation. The dissolution of the collision architecture induces the dissociation of both EDF1 and the ZNF598-GIGYF2-4EHP complex from the mRNA [40]. As 4EHP disengages from the 5’ cap, it no longer competes with eIF4E, thereby permitting the reinitiation of translation on the affected transcript. This hierarchical response system enables cellular machinery to dynamically modulate translation rates while maintaining translational fidelity, thus ensuring both qualitative and quantitative integrity of protein synthesis. Concurrently, ZAKα activates MAP kinase cascades of the ribotoxic stress response (RSR) [72], including p38 and JNK pathways [88,89], leading to cell cycle arrest or apoptotic responses depending on stress severity [90].

### 4.3. Integration of NMD, NGD, and NSD Mechanisms: GIGYF2’s Comprehensive Approach to mRNA Surveillance

When a translating ribosome encounters a premature termination codon, GIGYF2 becomes involved in the NMD through an RNA-dependent interaction with UPF1 and EJC components [91] (Figure 3C). UPF1, a key NMD factor with RNA helicase activity, initially associates with the terminating ribosome through interactions with release factors eRF1 and eRF3 [92]. While UPF1 phosphorylation by SMG1 is essential for NMD progression [93], the study shows this phosphorylation state does not affect the GIGYF2-UPF1 interaction [91]. The GIGYF2-4EHP complex functions primarily through translational repression of NMD targets with 3’ UTR introns [91,94,95,96]. These EJC-dependent NMD substrates characteristically display elevated ribosome densities. GIGYF2-EIF4E2 serves as a molecular brake that prevents excessive ribosome loading onto NMD targets, thereby reducing the likelihood of ribosome collisions and minimizing the number of ribosomes requiring rescue after SMG6-mediated transcript cleavage [97]. GIGYF2-4EHP regulates ribosomal occupancy of specific splicing isoforms like SRSF3, which contains an ultraconserved “poison cassette exon” with a PTC that triggers NMD. The SR splicing regulator encoded by SRSF3 autoregulates this exon’s inclusion, creating a feedback loop through alternative splicing and NMD to control its expression levels [98,99,100].

Additionally, GIGYF2 has been shown to inhibit the translation of non-stop decay (NSD) and no-go decay (NGD) substrates, with its inhibitory effect on NSD substrates being independent of ZNF598 [81]. The ribosome collisions activate the NGD pathway to degrade faulty mRNAs and initiate the RQC pathway to resolve stalled ribosomes, recycle ribosomal subunits, and degrade potentially harmful aberrant polypeptides via ZNF598-mediated ubiquitination of RPS10 [64,90]. Simultaneously, the EDF1-GIGYF2-4EHP complex inhibits translation initiation, preventing further ribosome loading onto problematic transcripts [71,101]. This provides a comprehensive cellular quality control mechanism that effectively suppresses the potential detrimental effects of aberrant polypeptides.

These surveillance mechanisms constitute a hierarchical quality control system that safeguards cellular proteostasis by preventing the accumulation of potentially toxic translation products. The dynamic interplay between transient and persistent ribosome collisions allows cells to calibrate their response according to translational stress severity, with GIGYF2 serving as a coordinator across multiple mRNA surveillance pathways. By operating at the critical nexus between translation and degradation, GIGYF2 ensures protein synthesis fidelity while linking translational stress to broader cellular responses.

## 5. GIGYF2-Mediated mRNA Degradation Mechanisms

GIGYF2 functions as a pivotal regulator of post-transcriptional gene expression by targeting specific mRNAs for translational repression and degradation. This multifunctional adaptor protein operates at the intersection of mRNA surveillance and decay pathways through diverse molecular mechanisms. Co-translational binding of GIGYF2 to mRNA marks transcripts with perturbed elongation for decay, a process that requires translation of the coding sequence (CDS) [15]. Through both cap-dependent inhibition and direct recruitment of degradation factors, GIGYF2 exerts precise control over mRNA fate in a context-dependent manner. We summarize four key mechanisms through which GIGYF2 mediates mRNA degradation, emphasizing its central role in RNA quality control and metabolism.

### 5.1. ZNF598-GIGYF2-4EHP Complex in Ribosome Collision-Mediated Decay

When ribosome collision occurs during translation of mRNAs encoding secreted and membrane-bound proteins or tubulin subunits, ZNF598 is recruited to the 40S-40S subunits of the collided ribosomes [64,80]. GIGYF2’s specific role in this pathway involves serving as a molecular scaffold that directly binds ZNF598 through its GYF domain via Pro-Pro-Gly-Φ motifs [15] (Figure 4A). The ribosome pausing and collision event serves as the essential initiating signal for GIGYF2-mediated translational repression and mRNA degradation [15].

Upon recruitment, GIGYF2 orchestrates the assembly of a degradation complex comprising CCR4-NOT4, 4EHP, and the DEAD-box RNA helicase DDX6 [15,25]. In this complex, GIGYF2 facilitates the functional coordination between different degradation factors: the CCR4-NOT4 complex mediates deadenylation of target mRNAs, while DDX6 promotes translational repression and facilitates decapping complex assembly [102,103,104]. This mechanism effectively integrates ribosome quality control with mRNA degradation through GIGYF2’s scaffolding function.

### 5.2. GIGYF2 as an RNA-Binding Protein in Direct mRNA Targeting

GIGYF2 functions as a sequence-specific RNA-binding protein (RBP) that directly recruits the CCR4-NOT complex to promote deadenylation of target mRNAs, accelerating their degradation through a mechanism independent of 4EHP [25,105] (Figure 4E). Five endogenous targets of GIGYF2-mediated silencing have been identified and validated: SVOPL, COL8A1, NPR3, CPM, and ECEL1 [25]. These targets were confirmed through RNA immunoprecipitation experiments demonstrating their direct association with GIGYF2, and their repression was shown to be independent of 4EHP binding [25]. GIGYF2 exhibits target selectivity with notable enrichment for mRNAs encoding proteins destined for the endoplasmic reticulum. Among the 23 endogenous RNAs identified as GIGYF2 targets, 15 out of 21 protein-coding transcripts encode transmembrane, GPI-anchored, or secreted proteins requiring ER-associated translation [25]. This spatial organization appears functionally relevant, as GIGYF2 localizes to the ER at steady-state [7], suggesting that GIGYF2-mediated repression may be spatially restricted to this cellular compartment. Notably, GIGYF2 can exert dual regulatory effects on mRNA stability as an RBP. The GIGYF2 fragment spanning amino acids 1-495 interacts with STAU1 mRNA and enhances its stability [12,106], demonstrating context-dependent regulatory functions.

### 5.3. GIGYF2’s Recruitment to Target Transcripts via RNA-Binding Proteins (RBPs)

GIGYF2 can be indirectly recruited to mRNAs through other RNA-binding proteins, particularly TTP and ZNF598, primarily facilitating 4EHP-dependent translational repression [25,38] (Figure 4C). GIGYF2’s role in this mechanism involves serving as a bridging factor that links sequence-specific RBPs to the translational repression machinery, particularly for inflammation-associated transcripts. The molecular basis of these interactions relies on GIGYF2’s GYF domains’ high affinity for polyproline stretches [107]. Both ZNF598 and TTP contain conserved polyproline stretches that bind to GIGYF2’s GYF domain, enabling integration into larger 4EHP-GIGYF2-RBP translation initiation inhibitory complexes [9,38]. ZNF598 and TTP exhibit partially overlapping mRNA-binding repertoires while maintaining distinct binding specificities. TTP recognizes adenylate-uridylate-rich elements (AREs) in 3’ UTRs through its two RNA-binding zinc fingers [108], whereas ZNF598 possesses C2H2-type zinc fingers with broader binding specificity for sequences in both coding and non-coding regions [109]. GIGYF2’s assembly function is critical for TTP-mediated regulation, as the formation of the 4EHP-GIGYF2-DDX6 complex is essential for TTP-mediated translational repression of AU-rich element-containing mRNAs [23]. GIGYF2’s recruitment of DDX6 serves as the critical molecular link between TTP’s ARE-binding activity and the translational repression machinery [23].

### 5.4. GIGYF2’s Role in miRNA-Mediated Silencing via TNRC6A Interaction

During miRNA-mediated silencing, GIGYF2 functions as a regulatory component that facilitates post-transcriptional gene regulation through direct interactions with TNRC6A in the miRNA-induced silencing complex (miRISC) [24,37] (Figure 4B). GIGYF2’s specific contribution involves facilitating the recruitment of 4EHP to Argonaute-miRNA complexes, thereby establishing a functional bridge between miRNA targeting machinery and translational repression factors [110]. Crystallographic analyses have elucidated the molecular basis of this interaction, revealing that GIGYF2’s GYF domain specifically recognizes and binds to a conserved PPGL motif within TNRC6A’s silencing domain [37,39]. TNRC6A serves as the primary scaffold protein within miRISC architecture [24,111], mediating CCR4-NOT complex recruitment via CNOT1 interactions [112,113,114]. CNOT1 directly binds DDX6, facilitating DDX6 localization to miRNA-targeted mRNAs [115].

### 5.5. Convergence on mRNA Degradation Pathways and P-Body Association

These four GIGYF2-mediated pathways converge on common mRNA degradation mechanisms involving deadenylation and decapping processes. While several GIGYF2-associated factors are known components of processing bodies (P-bodies), the extent of GIGYF2’s direct involvement in P-body dynamics requires further investigation [116,117,118,119]. Processing bodies represent discrete cytoplasmic foci that concentrate untranslating mRNAs with decay machinery, serving as sites where mRNAs undergo degradation, storage, or silencing [119,120]. The mRNA degradation process typically follows a sequential pathway: deadenylation through the CCR4-CAF1-NOT complex removes the poly(A) tail, followed by decapping via the DCP1/DCP2 complex and co-activators including Dhh1/RCK/p54, Pat1, and Lsm1-7 [121,122,123]. Subsequently, XRN1 mediates 5’-to-3’ exonucleolytic degradation [119,124]. Studies using Smy2 (GIGYF2 homolog) have provided insights into potential P-body associations, revealing binding to P-body components including Xrn1 exonuclease, Dcp2, seven Ccr4-NOT complex subunits, and Lsm proteins [7]. However, it should be noted that these binding partners are not exclusively localized to P-bodies and function in multiple cellular contexts.

### 5.6. Stress-Induced Regulation of GIGYF2 Activity

Under stress conditions, GIGYF2 undergoes regulatory phosphorylation at two evolutionarily conserved serine residues, S160 and S684, flanking the GYF domain [41] (Figure 4D). This phosphorylation is mediated by MAP kinase-activated protein kinase 2 (MK2), a downstream effector of the p38 MAPK pathway [42]. The resulting phosphorylation creates binding sites for 14-3-3 proteins, and this stress-induced 14-3-3 binding negatively modulates GIGYF2’s interactions with canonical binding partners, particularly those mediated by the GYF domain, thereby inhibiting GIGYF2-mediated transcriptional repression and mRNA degradation [41].

GIGYF2 orchestrates mRNA degradation through four distinct yet integrated pathways: coordinating degradation complexes at ribosome collision sites; directly binding target transcripts to recruit deadenylation factors; partnering with RNA-binding proteins to repress inflammation-associated mRNAs; and linking miRNA machinery to translational repression. These mechanisms converge on established mRNA degradation pathways involving deadenylation and decapping processes. Stress-induced phosphorylation enables 14-3-3 binding that modulates GIGYF2 activity, positioning it as a critical regulator that integrates diverse cellular signals to control transcript fate.

## 6. GIGYF2-Associated Pathological Conditions

The extensive involvement of GIGYF2 in multiple regulatory mechanisms spanning transcriptional control, translational repression, and mRNA degradation positions this protein as a significant coordinator of cellular homeostasis. The functional consequences of GIGYF2 activity extend beyond normal physiological processes to pathological states. The alterations in GIGYF2 expression, regulation, or protein–protein interactions have been documented across several disease contexts. The subsequent section examines the evidence linking GIGYF2 to specific pathological conditions, including metabolic diseases, vascular aging, viral infections, and neurodegenerative disorders, thereby elucidating the clinical implications of this molecular regulator.

### 6.1. GIGYF2 in Metabolic Diseases

In metabolic diseases, particularly hepatic insulin resistance and type 2 diabetes, GIGYF2 functions as an RNA-binding protein that enhances STAU1 mRNA stability [12,106] (Figure 5A). This upregulated STAU1 subsequently binds to and stabilizes PTEN transcripts [12,125], leading to increased PTEN protein expression that disrupts the PI3K/AKT signaling cascade. The consequent reduction in AKT phosphorylation at both Thr308 and Ser473 residues impairs downstream glucose metabolism and insulin sensitivity [126,127,128]. A pathological feedback loop is established where palmitic acid, abundant in obesity, induces GIGYF2 expression, triggering insulin resistance, which further exacerbates metabolic dysfunction [12].

### 6.2. GIGYF2 in Vascular Aging

In vascular aging, GIGYF2 promotes endothelial cell senescence and dysfunction through activation of the mTORC1-S6K1 signaling pathway [106] (Figure 5B). As an RNA-binding protein, GIGYF2 binds to and stabilizes STAU1 mRNA, leading to increased STAU1 expression. This upregulated STAU1 then binds to specific intron regions of LAMTOR4 mRNA, enhancing its stability and increasing LAMTOR4 expression [106]. Elevated LAMTOR4 facilitates the translocation of mTORC1 to lysosomes, activating the mTORC1-S6K1 signaling cascade [129,130,131,132]. This activation triggers eNOS uncoupling, resulting in decreased nitric oxide production and increased generation of reactive oxygen species, creating oxidative stress and endothelial dysfunction [133,134]. Consequently, senescent cells exhibit increased expression of senescence markers (p21, p16), decreased expression of anti-aging genes (SIRT1, SIRT6), and enhanced secretion of pro-inflammatory factors [106]. These cellular alterations contribute to vascular dysfunction through various mechanisms: overexpression of adhesion molecules (ICAM-1, VCAM-1), increased monocyte adhesion to endothelial cells, impaired endothelium-dependent vasodilation, and increased arterial stiffness [106]. Notably, GIGYF2 is hyper-expressed in senescent human endothelial cells and aged mouse aortas, while endothelial-specific GIGYF2 knockout provides protection against age-associated vascular dysfunction, highlighting its potential as a therapeutic target for vascular aging and age-related cardiovascular diseases.

### 6.3. GIGYF2 in Viral Infections

In viral infections, SARS-CoV-2 protein NSP2 directly interacts with human GIGYF2, enhancing the binding of GIGYF2 to 4EHP and partially increasing GIGYF2-mediated miRNA translation suppression [29,30,31] (Figure 5C). Notably, NSP2 displays selectivity, binding preferentially to the diffuse cytoplasmic form of GIGYF2 rather than its P-body-associated counterpart [31]. Studies have found that the NSP2-GIGYF2 interaction correlates with downstream effects on immune response, particularly the repression of Ifnb1 mRNA translation [30]. This results in reduced IFN-β levels, consequently diminishing the host’s antiviral immune defense mechanisms [135]. This interaction mechanism assumes greater significance in light of findings by Masood et al. demonstrating that upregulated type I interferon responses are associated with improved clinical outcomes in asymptomatic cases of COVID-19 [136,137]. Since SARS-CoV-2 NSP2 directly targets GIGYF2 to suppress IFN-β production, and production of type I interferons is vital to antiviral immunity as a host defense mechanism [135], disrupting the NSP2-GIGYF2 interaction could represent a promising therapeutic strategy. Zhou et al. (2020) established that SARS-CoV-2 shares 79.6% sequence identity with SARS-CoV and uses the same ACE2 receptor for cellular entry [138], providing further context for understanding viral pathogenesis. Given these findings, GIGYF2 warrants further investigation as a potential therapeutic target for SARS-CoV-2 infection, though additional studies are needed to validate this approach in appropriate experimental models and clinical settings.

### 6.4. GIGYF2 in Neurodegenerative Disorders

In neurodegenerative disorders, genetic studies have identified potential associations between GIGYF2 mutations and Parkinson’s disease, though the causative relationship remains debated due to inconsistent replication across different populations and the identification of mutations in both affected and unaffected individuals [21,38,139,140]. Animal model studies provide more compelling evidence for GIGYF2’s essential role in neurological function. Homozygous GIGYF2-null mice exhibit perinatal lethality due to feeding deficits, while heterozygous mice develop normally but progressively manifest age-related motor dysfunction, neurodegeneration, and disrupted IGF signaling [141], phenotypes reminiscent of human neurodegenerative conditions.

The molecular basis of GIGYF2’s neuroprotective function operates through two complementary mechanisms (Figure 5D). First, GIGYF2 directly interacts with neurological disease-associated proteins, including Atrophin-1 (implicated in dentatorubral pallidoluysian atrophy) and PQBP-1 (linked to intellectual disability) [7], potentially modulating their pathogenic activities. Second, GIGYF2 maintains protein homeostasis through its role in mRNA quality control. Specifically, the GIGYF2-4EHP complex prevents translation of nonsense-mediated decay (NMD) targets, thereby blocking production of potentially neurotoxic truncated proteins [91]. Consequently, GIGYF2 deficiency compromises this protective surveillance mechanism, allowing accumulation of aberrant proteins that may contribute to the neurodevelopmental and neuropsychiatric phenotypes observed in both mouse models and human patients [81].

The diverse pathologies associated with GIGYF2 dysregulation demonstrate its significance as a critical node in multiple disease networks. From metabolic disorders to vascular aging, viral infections, and neurodegenerative disorders, aberrant GIGYF2 activity contributes to disease progression through disruption of essential regulatory mechanisms. These findings collectively highlight GIGYF2 as a promising therapeutic target with broad clinical potential. Future research should focus on developing tissue-specific and context-dependent approaches to modulate GIGYF2 function, which may yield novel interventions for these challenging medical conditions while minimizing potential adverse effects on essential cellular processes.

## 7. Discussion

In various cellular contexts, GIGYF2 functions as an important regulator at the majority of stages of gene expression. At the transcriptional level, GIGYF2 is associated with transcriptional regulation through its interaction with VCP/p97 to extract ubiquitylated Rpb1 from stalled RNA polymerase II complexes during DNA damage response. In post-transcriptional processes, GIGYF2 is intricately involved in mRNA surveillance, translational repression, and mRNA degradation. Notably, under stress conditions, GIGYF2 also contributes to the subcellular localization of mRNAs to stress granules. As a translational repressor and regulator of RNA stability, GIGYF2 forms complexes through its multiple domains (N-terminal, GYF, and C-terminal) with multiple partners, including proteins and mRNAs (Table 2), to modulate gene expression in diverse cellular contexts. However, several questions regarding the precise molecular mechanisms of GIGYF2 function still remain unresolved.

### 7.1. Regulation of GIGYF2: An Underexplored Frontier

Despite the extensive characterization of GIGYF2’s downstream effects, the upstream regulatory mechanisms governing GIGYF2 expression, activity, and subcellular localization remain largely unexplored. Current knowledge is limited to stress-induced phosphorylation by MK2 kinase at serine residues S157 and S638, which recruits 14-3-3 proteins and modulates GYF domain-dependent interactions [41]. However, this represents only a fraction of the potential regulatory landscape. Several critical questions remain unanswered: What transcriptional and post-transcriptional mechanisms control GIGYF2 expression levels across different cell types and developmental stages? Are there additional post-translational modifications beyond phosphorylation that fine-tune GIGYF2 activity? How do different cellular stresses specifically modulate GIGYF2 function, and what are the upstream signaling cascades involved?

The tissue-specific expression patterns of GIGYF2, with particularly high levels in the liver, pancreas, brain, lung, kidney, and spleen [19], suggest the existence of tissue-specific regulatory programs that remain to be elucidated. Furthermore, the dynamic redistribution of GIGYF2 from ER/Golgi to stress granules under cellular stress [7] implies the existence of sophisticated trafficking mechanisms that are currently poorly understood. The identification of these regulatory networks will be crucial for understanding how GIGYF2 activity is coordinated with broader cellular programs and how dysregulation contributes to pathological conditions.

### 7.2. Nuclear Versus Cytoplasmic Functions: Complementary or Competitive Roles?

An intriguing aspect of GIGYF2 biology is its dual subcellular localization and the potential interplay between its nuclear and cytoplasmic functions. While GIGYF2 exhibits predominantly cytoplasmic localization where it regulates translation and mRNA degradation, a subset of GIGYF2 localizes to the nucleus where it participates in transcriptional regulation through the VCP/p97-mediated extraction pathway [49]. This raises fundamental questions about whether these nuclear and cytoplasmic roles represent complementary or potentially competitive functions.

The nuclear function of GIGYF2 in facilitating POLR2A extraction during transcriptional stress may indirectly influence the cytoplasmic mRNA landscape by affecting transcript abundance and quality. Conversely, the cytoplasmic functions of GIGYF2 in mRNA surveillance and degradation may influence which transcripts are available for nuclear export and subsequent processing. This bidirectional relationship suggests a potential coordination between nuclear quality control and cytoplasmic RNA metabolism that warrants systematic investigation.

Additionally, the identification of spliceosomal proteins (Prp8, Lsm4, U2AF65, and U2AF35) as potential GIGYF2 interaction partners [7] raises the possibility of previously uncharacterized nuclear functions beyond transcriptional stress responses. Unlike other GYF domain proteins that are primarily nuclear and involved in pre-mRNA splicing [11,142], GIGYF2’s Smy2-type GYF domain and predominantly cytoplasmic localization suggest unique functional adaptations that may bridge nuclear and cytoplasmic RNA processing events.

### 7.3. Disease-Associated Molecular Mechanisms: Current Limitations and Future Directions

While GIGYF2 has been implicated in diverse pathological conditions, including metabolic diseases, vascular aging, viral infections, and neurodegenerative disorders, the molecular mechanisms identified in disease contexts remain limited and often correlative rather than mechanistically definitive. This limitation stems from several factors that constrain our current understanding. First, most disease-associated studies have focused on single pathway analyses, failing to capture the multifaceted nature of GIGYF2 function. For instance, while the GIGYF2-STAU1-PTEN axis has been well-characterized in metabolic diseases [12], how this pathway interacts with GIGYF2’s roles in mRNA surveillance and translational control remains unclear. The potential crosstalk between different GIGYF2-mediated pathways in disease progression requires systematic investigation. Second, the temporal dynamics of GIGYF2 dysfunction in disease development are poorly understood. It remains unclear whether GIGYF2 alterations represent primary causative events, secondary adaptive responses, or consequences of disease progression. This distinction is crucial for determining whether GIGYF2 represents a viable therapeutic target or primarily a disease biomarker. Third, the tissue-specific and context-dependent nature of GIGYF2 function complicates the translation of findings across different disease models and human pathology. The same molecular pathway may have distinct consequences in different cellular environments, necessitating more nuanced approaches to understanding GIGYF2’s role in human disease.

### 7.4. Unresolved Mechanistic Questions and Future Perspectives

Several specific mechanistic questions require resolution to advance our understanding of GIGYF2 function. The precise binding interface between GIGYF2 and VCP/p97 remains incompletely characterized [49], limiting our understanding of how this interaction is regulated and potentially targeted therapeutically. Similarly, the molecular determinants governing GIGYF2’s recruitment to stress granules and its potential role in regulating P-body/stress granule dynamics during cellular stress require further investigation. The relationship between GIGYF2’s RNA-binding capacity and its protein-mediated recruitment to target mRNAs represents another area requiring clarification. While GIGYF2 can directly bind RNA through its N-terminal and C-terminal domains [25], the relative contributions of direct RNA binding versus protein-mediated recruitment in different cellular contexts remain unclear.

### 7.5. Conclusions and Therapeutic Implications

GIGYF2 serves as a versatile regulator across multiple levels of gene expression, spanning transcription, translation, mRNA surveillance, and degradation through diverse protein–protein and protein-RNA interactions. Its involvement in metabolic diseases, vascular aging, viral infections, and neurodegenerative disorders highlights its clinical significance. However, the current limitations in understanding GIGYF2 regulation, the relationship between its nuclear and cytoplasmic functions, and the molecular mechanisms underlying disease associations represent significant opportunities for future research.

Addressing these knowledge gaps will be essential for developing targeted therapeutic strategies that can modulate GIGYF2 function in a tissue-specific and context-dependent manner. The multifaceted nature of GIGYF2 function suggests that successful therapeutic interventions will require nuanced approaches that consider the complex interplay between its various cellular roles. Future research should prioritize systematic approaches to understand GIGYF2 regulation, comprehensive analyses of its subcellular functional organization, and mechanistic studies of its role in human disease pathogenesis.

## Figures and Tables

**Figure 1 cells-14-01032-f001:**
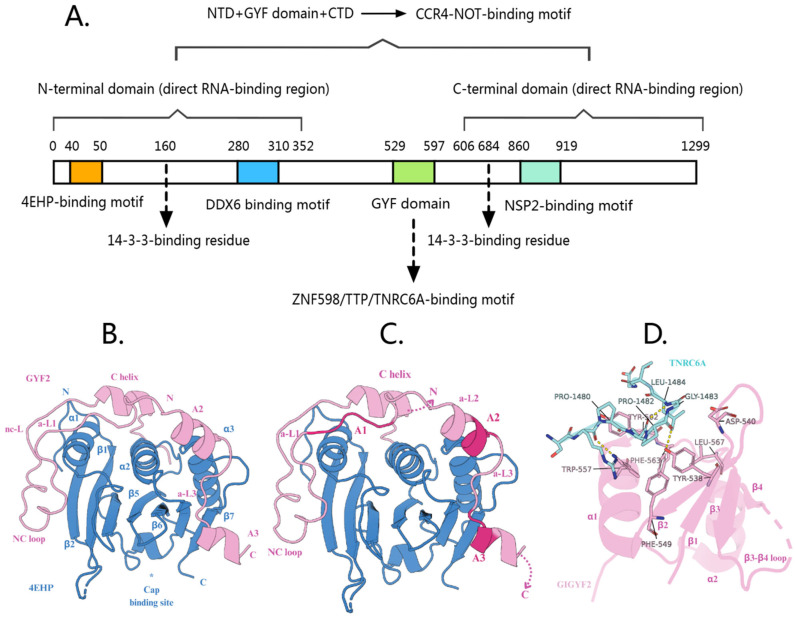
(**A**) The functional domains of GIGYF2 protein are involved in gene expression regulation. (**B**) Crystal structure of the 4EHP-GIGYF2 complex. 4EHP is shown in blue and GIGYF2 in pink, illustrating the extensive binding interface and stable heterodimeric complex formation. Asterisks (*) indicate residues that are shown without their side chains for clarity. (C) Schematic representation of the tripartite binding mechanism between 4EHP (blue) and GIGYF2 (pink). The diagram shows three distinct binding regions: canonical motif (**C**) at the dorsal surface, non-canonical motif (NC) at the lateral surface, and auxiliary sequences (A1–A3) providing 4EHP-specific contacts. Auxiliary motifs are interconnected by linkers a-L1, a-L2, and a-L3, with the invariant PLAL motif in A1. This cooperative binding mechanism confers high affinity and specificity for 4EHP over eIF4E. (**D**) Crystal structure of GIGYF2 GYF domain (pink cartoon) in complex with TNRC6A PRS peptide (blue sticks). The peptide binds to a conserved hydrophobic groove formed by aromatic residues on the GYF domain surface. Yellow dashed lines indicate hydrogen bonds. This interaction facilitates recruitment of the 4EHP-GIGYF2 complex to miRISC for translational repression.

**Figure 2 cells-14-01032-f002:**
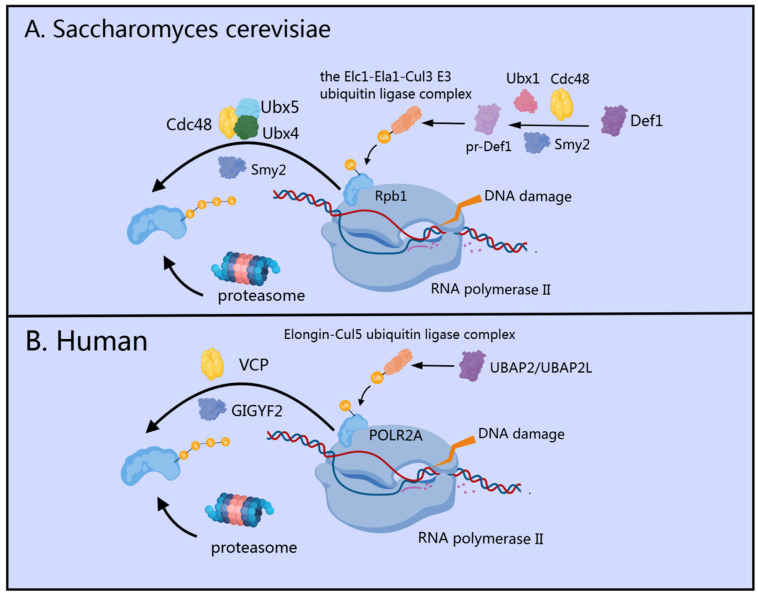
Role of GIGYF2/Smy2 in the “Last Resort” Pathway During Transcription Stress Response. (**A**) In *Saccharomyces cerevisiae*, Smy2 collaborates with Ubx1 and Cdc48 to facilitate proteasome-mediated processing of Def1, recruiting the Elc1-Ela1-Cul3 ligase complex to monoubiquitylated Rpb1. Smy2 then works with Cdc48 and its co-factors Ubx4/5 to extract polyubiquitylated Rpb1 from damage-stalled RNAPII complexes for proteasomal degradation. (**B**) In human cells, GIGYF2 interacts with VCP (mammalian homolog of Cdc48) to facilitate extraction and proteasomal degradation of ubiquitylated POLR2A from stalled transcription complexes following UBAP2/UBAP2L-mediated recruitment of the Elongin-Cul5 ubiquitin ligase complex.

**Figure 3 cells-14-01032-f003:**
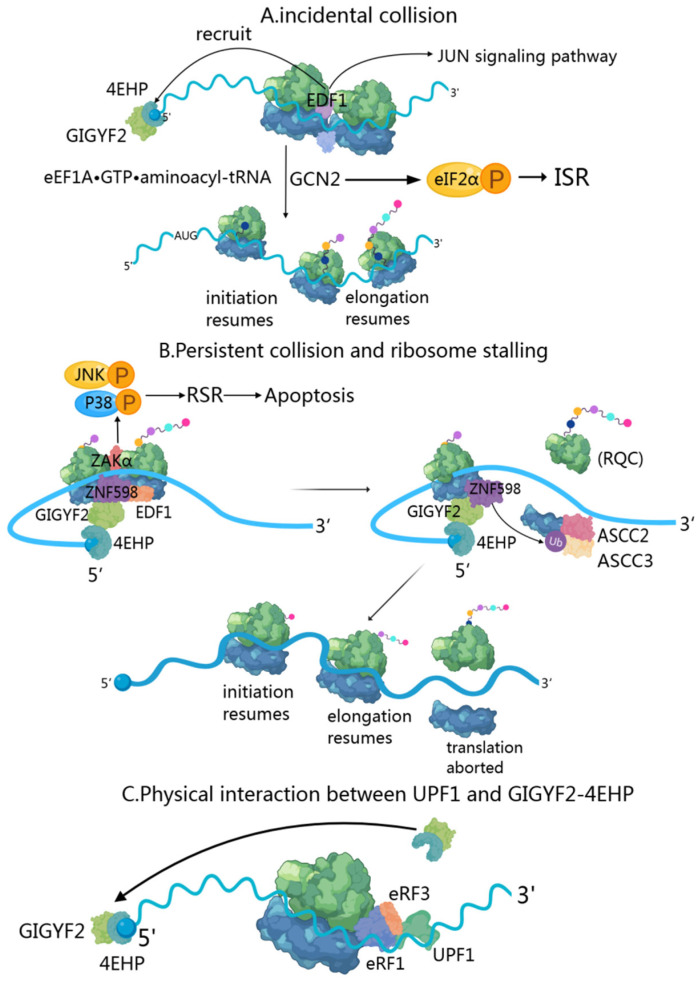
GIGYF2’s role in mRNA surveillance. (**A**) Transient ribosome collision. EDF1 rapidly binds to collided ribosomes and recruits the GIGYF2-4EHP complex through RACK1-dependent mechanisms. GIGYF2-4EHP inhibits translation by competing with eIF4E for 5’ cap binding, reducing ribosome density. GCN2 senses collisions and activates the ISR pathway via eIF2α phosphorylation. (**B**) Persistent collision and ribosome stalling. ZNF598 recognizes collided di-ribosomes and ubiquitinates eS10, triggering ASCC-mediated ribosome dissociation. The GIGYF2-4EHP complex continues translation inhibition while RQC machinery processes incomplete peptides. ZAKα activates RSR pathways (p38/JNK), leading to stress responses. (**C**) Nonsense-mediated decay regulation. GIGYF2 interacts with UPF1 and EJC components at premature termination codons. The GIGYF2-4EHP complex prevents excessive ribosome loading on NMD targets, reducing collision likelihood. GIGYF2 also regulates NGD and NSD substrates, providing comprehensive quality control against aberrant mRNAs.

**Figure 4 cells-14-01032-f004:**
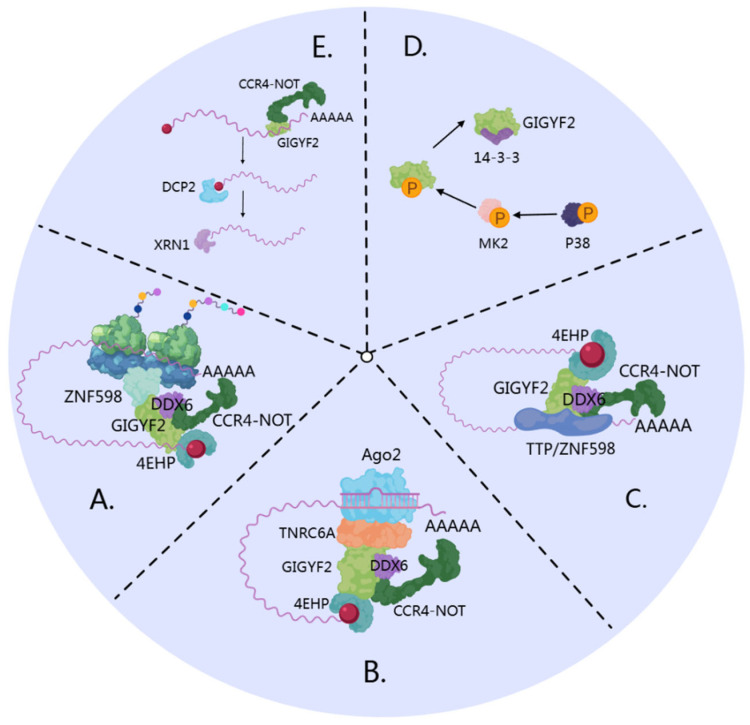
GIGYF2-mediated mRNA decay pathways converging on P-bodies. (**A**) Ribosome collision-mediated decay: GIGYF2 serves as a molecular scaffold binding ZNF598 and orchestrating assembly of the CCR4-NOT4-4EHP-DDX6 degradation complex. GIGYF2-4EHP inhibits translation initiation by competing with eIF4E for 5’ cap binding. (**B**) miRNA-mediated silencing: GIGYF2 bridges miRNA targeting machinery (AGO2-miRNA-TNRC6A) and translational repression factors (4EHP) by binding TNRC6A through its GYF domain, facilitating target mRNA silencing. (**C**) TTP/ZNF598-mediated recruitment: GIGYF2 links sequence-specific RNA-binding proteins (TTP and ZNF598) to the translational repression machinery through GYF domain interactions, enabling 4EHP-mediated repression of inflammation-associated mRNAs. (**D**) Stress-induced regulation: p38 MAPK-activated MK2 phosphorylates GIGYF2 (S157, S638), creating 14-3-3 binding sites that negatively modulate GIGYF2’s interactions with GYF domain partners, inhibiting GIGYF2-mediated mRNA regulation. (**E**) Direct RNA-binding function: GIGYF2 directly binds target mRNAs (especially ER-destined transcripts) and recruits the CCR4-NOT complex for deadenylation, promoting mRNA degradation independent of 4EHP. All pathways converge through conventional decay involving DCP2 decapping and XRN1 exonucleolytic degradation.

**Figure 5 cells-14-01032-f005:**
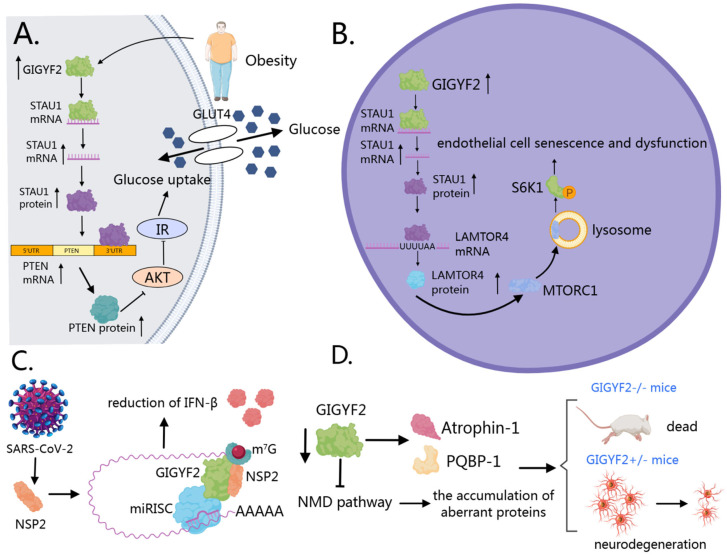
GIGYF2’s roles in various pathological conditions. (**A**) Metabolic diseases: GIGYF2 stabilizes STAU1 mRNA, leading to increased STAU1 expression that enhances PTEN mRNA stability. Elevated PTEN disrupts PI3K/AKT signaling, reducing AKT phosphorylation and impairing glucose metabolism and insulin sensitivity. Palmitic acid induces GIGYF2 expression, creating a pathological feedback loop that exacerbates metabolic dysfunction. (**B**) Vascular aging: GIGYF2 promotes endothelial senescence by stabilizing STAU1 mRNA, which enhances LAMTOR4 mRNA stability. Elevated LAMTOR4 activates mTORC1-S6K1 signaling, triggering eNOS uncoupling and reducing nitric oxide production while increasing reactive oxygen species, leading to endothelial dysfunction and vascular aging. (**C**) Viral infections: SARS-CoV-2 NSP2 directly interacts with GIGYF2, enhancing GIGYF2-4EHP binding and increasing miRNA-mediated translation suppression. This interaction represses Ifnb1 mRNA translation, reducing IFN-β levels and compromising the host’s antiviral immune response. (**D**) Neurodegenerative disorders: GIGYF2 maintains neuronal proteostasis through direct interactions with disease-associated proteins (Atrophin-1, PQBP-1) and mRNA quality control via the GIGYF2-4EHP complex. GIGYF2 deficiency allows accumulation of aberrant proteins, contributing to neurodevelopmental and neuropsychiatric phenotypes.

**Table 2 cells-14-01032-t002:** GIGYF2’s protein and RNA interactions in gene expression regulation.

Interacting Partner	Interaction Domain	References	Functional Consequence of the Interaction	References
4EHP	residues 40–50	[28]	Forms a complex that inhibits translation initiation by competing with eIF4E for 5’ cap binding	[20]
DDX6	residues 280–310	[23]	Essential for TTP-mediated translational repression of AU-rich mRNAs	[117]
ZNF598	GYF domain (residues 529–597)	[38]	Recruits GIGYF2 to collided ribosomes during quality control of translation	[40]
TNRC6A		[37]	Connects GIGYF2-4EHP complex to miRNA-regulated transcripts	[37]
TTP		[38]	Recruits GIGYF2-4EHP complex to ARE-containing mRNAs (inflammation-associated transcripts)	[38]
NSP2	residues 860–919	[30]	Enhances the interaction between GIGYF2 and 4EHP	[31]
CCR4-NOT	multiple domains	[25]	Promotes deadenylation of target mRNAs	[25]
14-3-3 proteins	phosphorylated serine residues S157 and S638	[41]	Negatively modulates GIGYF2’s interactions with GYF domain-binding partners during stress conditions	[41]
UPF1	unknown		Regulates ribosomal occupancy of mRNA containing premature termination codons	[91]
VCP/p97	unknown		Promotes extraction and proteasomal degradation of ubiquitylated Rpb1 from stalled transcription complexes during DNA damage response	[49]
EDF1	unknown		Recruits and stabilizes GIGYF2-4EHP complex to ribosome collision sites	[40]
STAU1 mRNA	residues 1–495	[12,106]	Enhances STAU1 mRNA stability	[12]
SVOPL mRNA	N-terminal domain (residues 1–532) and C-terminal domain (residues 606–1299)	[25]	Direct engagement with endogenous mRNAs for translational regulation and degradation	[25]
COL8A1 mRNA				
NPR3 mRNA				
CPM mRNA				
ECEL1 mRNA				

## Data Availability

No new data were created or analyzed in this study.

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
