# Peer review of "GIGYF2: A Multifunctional Regulator at the Crossroads of Gene Expression, mRNA Surveillance, and Human Disease"

_cells, 2025, doi:10.3390/cells14131032_

Round 1
Reviewer 1 Report
Comments and Suggestions for Authors
Please find the comments to the authors in the attached file.

Author Response
Comments 1: Page 3, second paragraph. The text is a bit unclear regarding the multiple regions present in the 4EHP-binding motif. The authors should clarify that the motif as canonical, noncanonical and in addition auxiliary sequences that mediate the binding of GIGYF2 to 4EHP. Maybe a structural figure (additional panels in figure 1) accompanying the description of the binding modes between GIGYF2 and its partners (4EHP, DDX6, GYF domain binding proteins) would help the reader to follow the detailed information described in the text.
Response 1:
The 4EHP-binding region (4EHP-BR) of GIGYF2 employs a tripartite binding mechanism consisting of three distinct sequence elements that work cooperatively to achieve high-affinity and specific binding to 4EHP (Peter et al., 2017) (Table 1) (Figure 1B). First, the canonical motif (YXYX4LΦ consensus sequence, residues 41-49) forms the primary interaction by binding to the dorsal surface of 4EHP in a helical conformation, similar to how 4E-BPs bind to eIF4E. Second, the noncanonical motif (located ~12 residues C-terminal to the canonical motif) engages the lateral surface of 4EHP through hydrophobic interactions, providing additional binding stability. Third, and most importantly for 4EHP specificity, the auxiliary sequences (A1-A3) extend the binding interface and contact 4EHP-specific residues that are not conserved in eIF4E.
The auxiliary sequences can be subdivided into three functional elements: (1) Auxiliary motif 1 (A1), which contains the invariant PLAL motif and is connected to the noncanonical motif via auxiliary linker 1 (a-L1); (2) Auxiliary motif 2 (A2), which forms auxiliary helix α2; and (3) Auxiliary motif 3 (A3), which adopts auxiliary helix α3 and is positioned near the 4EHP cap-binding pocket (Figure 1C). These auxiliary motifs are interconnected by conserved linkers a-L2 and a-L3, with the a-L3 linker containing a critical VNS sequence that interacts with 4EHP-specific residues (Peter et al., 2017). Importantly, the auxiliary sequences increase GIGYF2's affinity for 4EHP by 30-40 fold compared to peptides lacking these sequences, and they interact specifically with 4EHP residues (R103, R140, E149) that differ from eIF4E, thereby conferring selectivity for 4EHP over eIF4E.
(Thank you for this important observation. We acknowledge that our structural analysis is limited by the current availability of structural data in major databases. While crystal structures exist for GIGYF2-4EHP and GIGYF2-TNRC6A complexes, high-resolution structural data for the human GIGYF2-DDX6 complex are not yet available, despite strong biochemical evidence for this interaction. We have ensured our analysis appropriately reflects these data limitations.)
Comments 2:Page 4, lines 14-15 – the authors should clarify in the text that NOT1 interacts with a DDX6 in a surface adjacent to the FDF pocket of DDX6 engaged by other partners of the RNA helicase and that depending on this binding partner the interaction with NOT1 may or not be favored. Like in the case of the 4E-T-DDX6 interaction, binding of GIGYF to DDX6 could permit the assembly of the GIGYF-DDX6-NOT1 complex.
Response 2:
GIGYF2 interacts with the RNA-dependent ATPase DDX6/Me31B through a conserved Me31B/DDX6-binding motif (MBM) to mediate translational repression (Mathys et al., 2014; Ozgur et al., 2015; Peter et al., 2019). Structurally, the MBM employs a bipartite binding mechanism: the N-terminal PEW (Pro-Glu-Trp) sequence forms a short coil, with the tryptophan residue (W349 in Drosophila GIGYF, equivalent to W221 in 4E-T) inserting into the W pocket between helices α10 and α11 of Me31B (Table 1). The PEW region is connected via a flexible linker to a C-terminal β-hairpin structure containing a unique "split" FDxâ‚„F motif (F361, D362, and F367 in Drosophila GIGYF) that engages the FDF pocket of Me31B.
This bipartite binding strategy distinguishes GIGYF from other DDX6-interacting proteins by combining structural elements reminiscent of the 4E-T PEW motif with its specialized "split" FDxâ‚„F configuration. Critically, the molecular basis for selective ternary complex formation lies in the differential binding partner requirements. NOT1, the scaffold protein of the CCR4-NOT deadenylase complex, binds to a distinct surface on the DDX6 RecA2 domain that is spatially adjacent to, but non-overlapping with, the FDF pocket utilized by other DDX6 partners (Peter et al., 2019).
The capacity for ternary complex assembly is determined by the electrostatic compatibility between binding partners and NOT1. Specifically, EDC3 and PatL1 contain negatively charged residues N-terminal to their FDF motifs, which generate unfavorable electrostatic repulsions with NOT1, precluding ternary complex formation. In contrast, both 4E-T and GIGYF feature polar rather than negatively charged residues preceding their respective binding motifs (IEL and FDxâ‚„F), thereby permitting the formation of stable ternary complexes: 4E-T-DDX6-NOT1 and GIGYF-DDX6-NOT1 (Peter et al., 2019). This structural compatibility enables GIGYF to orchestrate translational repression of target mRNAs through coordinated recruitment of both the DDX6 helicase and the CCR4-NOT deadenylation machinery.
Comments 3:GIGYF2 in P-bodies. Although GIGYF2 binds to multiple proteins that can be present in P-bodies, GIGYF2 subcellular distribution does not match with P-bodies in the majority of the cell types examined to date. I suggest that the idea that GIGYF2 might be present in P-bodies and regulates P-bodies dynamics is tone down, as no good evidence is
currently present in the literature. These partners of GIGYF2 are also not only present in P-bodies.
Response 3:
These four GIGYF2-mediated pathways converge on common mRNA degradation mechanisms involving deadenylation and decapping processes. While several GIGYF2-associated factors are known components of processing bodies (P-bodies), the extent of GIGYF2's direct involvement in P-body dynamics requires further investigation (Adjibade & Mazroui, 2014; Eulalio et al., 2007; Liu et al., 2005; Majerciak et al., 2023). Processing bodies represent discrete cytoplasmic foci that concentrate untranslating mRNAs with decay machinery, serving as sites where mRNAs undergo degradation, storage, or silencing (Eulalio et al., 2007; Parker & Sheth, 2007). The mRNA degradation process typically follows a sequential pathway: deadenylation through the CCR4-CAF1-NOT complex removes the poly(A) tail, followed by decapping via the DCP1/DCP2 complex and co-activators including Dhh1/RCK/p54, Pat1, and Lsm1-7 (Ingelfinger et al., 2002; Sheth & Parker, 2003; van Dijk et al., 2002). Subsequently, XRN1 mediates 5'-to-3' exonucleolytic degradation (Eulalio et al., 2007; Orban & Izaurralde, 2005). Studies using Smy2 (GIGYF2 homolog) have provided insights into potential P-body associations, revealing binding to P-body components including Xrn1 exonuclease, Dcp2, seven Ccr4-NOT complex subunits, and Lsm proteins (Ash et al., 2010). However, it should be noted that these binding partners are not exclusively localized to P-bodies and function in multiple cellular contexts.
Comments 4: Page 20, second paragraph. The discussion regarding the role of GIGYF2 protein in COPII coated vesicles seems very speculative. Besides the possible interaction of the Smy2-Gyf domain in some Sec complexes in yeast, there is no additional data supporting the role of GIGYF2 in vesicle formation, movement along microtubules and even association with
this type of vesicles. I suggest this hypothesis is removed if the authors cannot support this possibility further. The same remark concerning the role of GIGYF2 in splicing (third point of the discussion): only possible interaction with Smy2 GYF domain is known. Other interesting discussing points that could be mentioned instead: regulation of GIGYF2 proteins remains largely unexplored, nuclear vs cytoplasmic roles are complementary or not, molecular mechanism identified in disease are currently limited…
Response 4:
7.1. Regulation of GIGYF2: An Underexplored Frontier
Despite the extensive characterization of GIGYF2's downstream effects, the upstream regulatory mechanisms governing GIGYF2 expression, activity, and subcellular localization remain largely unexplored. Current knowledge is limited to stress-induced phosphorylation by MK2 kinase at serine residues S157 and S638, which recruits 14-3-3 proteins and modulates GYF domain-dependent interactions (Nordgaard et al., 2021). However, this represents only a fraction of the potential regulatory landscape. Several critical questions remain unanswered: What transcriptional and post-transcriptional mechanisms control GIGYF2 expression levels across different cell types and developmental stages? Are there additional post-translational modifications beyond phosphorylation that fine-tune GIGYF2 activity? How do different cellular stresses specifically modulate GIGYF2 function, and what are the upstream signaling cascades involved?
The tissue-specific expression patterns of GIGYF2, with particularly high levels in liver, pancreas, brain, lung, kidney, and spleen (Higashi et al., 2010), suggest the existence of tissue-specific regulatory programs that remain to be elucidated. Furthermore, the dynamic redistribution of GIGYF2 from ER/Golgi to stress granules under cellular stress (Ash et al., 2010) implies the existence of sophisticated trafficking mechanisms that are currently poorly understood. The identification of these regulatory networks will be crucial for understanding how GIGYF2 activity is coordinated with broader cellular programs and how dysregulation contributes to pathological conditions.
7.2. Nuclear versus Cytoplasmic Functions: Complementary or Competitive Roles?
An intriguing aspect of GIGYF2 biology is its dual subcellular localization and the potential interplay between its nuclear and cytoplasmic functions. While GIGYF2 exhibits predominantly cytoplasmic localization where it regulates translation and mRNA degradation, a subset of GIGYF2 localizes to the nucleus where it participates in transcriptional regulation through the VCP/p97-mediated extraction pathway (Lehner et al., 2022). This raises fundamental questions about whether these nuclear and cytoplasmic roles represent complementary or potentially competitive functions.
The nuclear function of GIGYF2 in facilitating POLR2A extraction during transcriptional stress may indirectly influence the cytoplasmic mRNA landscape by affecting transcript abundance and quality. Conversely, the cytoplasmic functions of GIGYF2 in mRNA surveillance and degradation may influence which transcripts are available for nuclear export and subsequent processing. This bidirectional relationship suggests a potential coordination between nuclear quality control and cytoplasmic RNA metabolism that warrants systematic investigation.
Additionally, the identification of spliceosomal proteins (Prp8, Lsm4, U2AF65, U2AF35) as potential GIGYF2 interaction partners (Ash et al., 2010) raises the possibility of previously uncharacterized nuclear functions beyond transcriptional stress responses. Unlike other GYF domain proteins that are primarily nuclear and involved in pre-mRNA splicing (Bialkowska & Kurlandzka, 2002; Kofler et al., 2004), GIGYF2's Smy2-type GYF domain and predominantly cytoplasmic localization suggest unique functional adaptations that may bridge nuclear and cytoplasmic RNA processing events.
7.3. Disease-Associated Molecular Mechanisms: Current Limitations and Future Directions
While GIGYF2 has been implicated in diverse pathological conditions including metabolic diseases, vascular aging, viral infections, and neurodegenerative disorders, the molecular mechanisms identified in disease contexts remain limited and often correlative rather than mechanistically definitive. This limitation stems from several factors that constrain our current understanding. First, most disease-associated studies have focused on single pathway analyses, failing to capture the multifaceted nature of GIGYF2 function. For instance, while the GIGYF2-STAU1-PTEN axis has been well-characterized in metabolic diseases (Lv et al., 2024), how this pathway interacts with GIGYF2's roles in mRNA surveillance and translational control remains unclear. The potential crosstalk between different GIGYF2-mediated pathways in disease progression requires systematic investigation. Second, the temporal dynamics of GIGYF2 dysfunction in disease development are poorly understood. It remains unclear whether GIGYF2 alterations represent primary causative events, secondary adaptive responses, or consequences of disease progression. This distinction is crucial for determining whether GIGYF2 represents a viable therapeutic target or primarily a disease biomarker. Third, the tissue-specific and context-dependent nature of GIGYF2 function complicates the translation of findings across different disease models and human pathology. The same molecular pathway may have distinct consequences in different cellular environments, necessitating more nuanced approaches to understanding GIGYF2's role in human disease.
7.4. Unresolved Mechanistic Questions and Future Perspectives
Several specific mechanistic questions require resolution to advance our understanding of GIGYF2 function. The precise binding interface between GIGYF2 and VCP/p97 remains incompletely characterized (Lehner et al., 2022), limiting our understanding of how this interaction is regulated and potentially targeted therapeutically. Similarly, the molecular determinants governing GIGYF2's recruitment to stress granules and its potential role in regulating P-body/stress granule dynamics during cellular stress require further investigation. The relationship between GIGYF2's RNA-binding capacity and its protein-mediated recruitment to target mRNAs represents another area requiring clarification. While GIGYF2 can directly bind RNA through its N-terminal and C-terminal domains (Amaya Ramirez et al., 2018), the relative contributions of direct RNA binding versus protein-mediated recruitment in different cellular contexts remain unclear.
7.5. Conclusions and Therapeutic Implications
GIGYF2 serves as a versatile regulator across multiple levels of gene expression, spanning transcription, translation, mRNA surveillance, and degradation through diverse protein-protein and protein-RNA interactions. Its involvement in metabolic diseases, vascular aging, viral infections, and neurodegenerative disorders highlights its clinical significance. However, the current limitations in understanding GIGYF2 regulation, the relationship between its nuclear and cytoplasmic functions, and the molecular mechanisms underlying disease associations represent significant opportunities for future research.
Addressing these knowledge gaps will be essential for developing targeted therapeutic strategies that can modulate GIGYF2 function in a tissue-specific and context-dependent manner. The multifaceted nature of GIGYF2 function suggests that successful therapeutic interventions will require nuanced approaches that consider the complex interplay between its various cellular roles. Future research should prioritize systematic approaches to understand GIGYF2 regulation, comprehensive analyses of its subcellular functional organization, and mechanistic studies of its role in human disease pathogenesis.
Comments/Response 5:
We sincerely appreciate your thoughtful comments and valuable suggestions. All other issues you raised have been carefully addressed, and we have made the corresponding revisions in the manuscript accordingly. Thank you for your time and effort in reviewing our work.
Reviewer 2 Report
Comments and Suggestions for Authors
This is a rather complete review of the protein GIGYF2 addressing its many interactions with various protein partners and their involvement in different cellular processes. This reviewer has two major concerns that hopefully the authors can address. The first is the molar amount of GiGYF2 in the cell if there are so many possible binding partners (is there a competition between different binding partners, might this be influenced by post-translational modifications?). The second concern is about the cellular location of GIGYF2. Is it truly mostly ER associated (which might explain a predominant effect on secreted proteins) or is there some specific mechanism to allow for nuclear localization (nuclear localization signal, nuclear export signal)?
Additional concerns
- Page 3, line 20 – it would be helpful to show the a-L2 and a-L3 segments in Figure 1.
- Page 4, line 37 – are TNRC6A, TTP and ZNF598 constant binding partners of GIGYF2 or transient? If transient, under what conditions?
- Page 4, line 44 – is the targeting of GIGYF2 to mRNAs through its RNA binding region or is it through the recognition of the peptide emerging from the ribosome?
- Page 9, line 26 – The complex should be eEF1A•GTP•aminoacyl-tRNA (include GTP in the figure as well).
- Figure 3 – GIGYF2 is not shown for panels A and B. Is there a reason for this?
- Page 13, line 22 – Is there a common theme for the proteins SVOPL, COLSA1, NPR3, CPM and ECEL1 (i.e. are they all secreted?)?
- Figure 4 – for some panels, the role of GIGYF2 is not clear.
- Figure 5 – this figure needs a more complete figure legend. Secondly, panel D is not clear.
- Latin terms or names should be in italics.
Author Response
Comments 1:
Page 4, line 37 – are TNRC6A, TTP and ZNF598 constant binding partners of GIGYF2 or transient? If transient, under what conditions?
Response 1:
Importantly, these interactions are predominantly transient and context-dependent rather than constitutive associations. TNRC6A-GIGYF2 interactions are specifically recruited during miRNA-mediated gene silencing when target mRNAs are bound by the miRISC complex (Schopp et al., 2017). TTP-GIGYF2 associations are primarily induced under inflammatory conditions when AU-rich element-containing mRNAs require translational repression (Tollenaere et al., 2019). ZNF598-GIGYF2 interactions are triggered during ribosome collision events and translational quality control responses (Juszkiewicz, Slodkowicz, et al., 2020; Weber et al., 2020). Additionally, these GYF domain-dependent interactions can be negatively modulated by stress-induced GIGYF2 phosphorylation, which recruits 14-3-3 proteins and disrupts binding partner associations (Nordgaard et al., 2021).
Comments 2:
Page 4, line 44 – is the targeting of GIGYF2 to mRNAs through its RNA binding region or is it through the recognition of the peptide emerging from the ribosome?
Response 2:
Notably, GIGYF2 targeting to mRNAs occurs through dual mechanisms: indirect recruitment via protein-protein interactions with RNA-binding partners (such as TTP recognizing AU-rich elements or TNRC6A in miRISC complexes), and direct RNA binding through its intrinsic RNA-binding regions located in both the N-terminal (residues 1-532) and C-terminal (residues 606-1299) domains (Amaya Ramirez et al., 2018). In the context of ribosome collision events, GIGYF2 recruitment is primarily mediated through ZNF598 recognition of collided ribosome surfaces rather than through direct RNA binding or nascent peptide recognition (Juszkiewicz, Slodkowicz, et al., 2020; Weber et al., 2020). This molecular architecture enables GIGYF proteins to participate in context-specific translational control through a common binding mechanism with partner-specific variations.
Comments 3:
Is there a common theme for the proteins SVOPL, COLSA1, NPR3, CPM and ECEL1 (i.e. are they all secreted?)?
Response 3:
Five endogenous targets of GIGYF2-mediated silencing have been identified and validated: SVOPL, COL8A1, NPR3, CPM, and ECEL1 (Amaya Ramirez et al., 2018). These targets were confirmed through RNA immunoprecipitation experiments demonstrating their direct association with GIGYF2, and their repression was shown to be independent of 4EHP binding (Amaya Ramirez et al., 2018). GIGYF2 exhibits target selectivity with notable enrichment for mRNAs encoding proteins destined for the endoplasmic reticulum. Among the 23 endogenous RNAs identified as GIGYF2 targets, 15 out of 21 protein-coding transcripts encode transmembrane, GPI-anchored, or secreted proteins requiring ER-associated translation (Amaya Ramirez et al., 2018). This spatial organization appears functionally relevant, as GIGYF2 localizes to the ER at steady-state (Ash et al., 2010), suggesting that GIGYF2-mediated repression may be spatially restricted to this cellular compartment.
Comments 4:
Figure 3 – GIGYF2 is not shown for panels A and B. Is there a reason for this?
Response 4:
We sincerely appreciate your comments regarding the figures. We have carefully reviewed and corrected all the identified errors in the figures. The revised figures have been updated in the manuscript accordingly.
Comments/Response 5:
We sincerely appreciate your thoughtful comments and valuable suggestions. All other issues you raised have been carefully addressed, and we have made the corresponding revisions in the manuscript accordingly. Thank you for your time and effort in reviewing our work.